# The genetics of extrinsic postzygotic selection in a migratory divide between subspecies of the Swainson's thrush

Hannah C. Justen [1] ✉, Stephanie A. Blain [2] & Kira E. Delmore[2]

Extrinsic postzygotic isolation, where hybrids experience reductions in fitness due to a mismatch with their environment, is central to speciation. Knowledge of genetic variants that underlie extrinsic isolation is crucial for understanding the early stages of speciation. Differences in seasonal migration are strong candidates for extrinsic isolation (e.g., if hybrids take intermediate and inferior routes compared to pure forms). Here, we used a hybrid zone between two subspecies of the songbird Swainson's thrush (*Catharus ustulatus*) with different migratory routes and tests for viability selection (locus-specific changes in interspecific heterozygosity and ancestry mismatch across age classes) to gain insight into the genetic basis of extrinsic isolation. Using data from over 900 individuals we find strong evidence for viability selection on both interspecific heterozygosity and ancestry mismatch at loci linked to migration. Much of this selection was dependent on genome-wide ancestry; as expected, a subset of hybrids exhibited reduced viability, but remarkably, another subset appears to fill an unoccupied fitness peak within the species, exhibiting higher viability than even parental forms. Many of the variants that influence hybrid viability appear to occur in structural variants, including a putative pericentric inversion. Our study emphasizes the importance of epistatic interactions and structural variants in speciation.

Knowledge of genetic variants that underlie selection against hybrids (i.e., postzygotic isolation) is critical for understanding speciation[1]. Hybrid zones are common in nature, suggesting that prezygotic isolating barriers are often incomplete[2–5], and some form of selection against hybrids must help maintain reproductive isolation in these systems[6–8]. Most work on the genetic variants that reduce hybrid fitness has focused on intrinsic postzygotic isolation (i.e., isolation that arises from genetic incompatibilities between parental genomes in hybrid [admixed] genomes and leads to hybrid sterility or inviabiltiy[9–11]). Few studies have focused on the genetic variants that underlie extrinsic postzygotic isolation (i.e., isolation that derives from divergent natural selection between parental environments[12,13]), despite its potential to act early in divergence[11,14] and explain several

patterns in natural populations[8,15]. Our recent ability to obtain large genomic datasets from natural populations will allow us to fill this knowledge gap.

Most work on the genetics of speciation in natural populations uses patterns of genomic variation to identify signatures of selection in the genome (e.g., scans for elevated genomic divergence and comparisons of synonymous and non-synonymous substitutions[16]). Unfortunately, many of these approaches provide only partial, and sometimes biased, evidence for selection. For example, many of these approaches cannot distinguish between different kinds of selection or test for the effects of additional variables (e.g., age or sex[17]). These approaches may also be affected by processes unrelated to selection (e.g., genomic features like centromeres and other areas of reduced

¹Neurobiology, Physiology and Behavior Department, University of California, Davis, CA, USA. ²Ecology and Evolutionary Biology Department, Columbia University, New York City, USANY. ✉e-mail: hjusten@ucdavis.edu

recombination that extend the effects of linked background selection[18–20]), and be biased towards alleles of large effect[21,22] and/or specific time scales[23]. Tests for viability selection (locus-specific changes in allele frequency across age classes) can overcome many of these challenges, allowing researchers to observe the contemporary effects of genotypes on survival by identifying loci that exhibit changes in allele frequency across age classes[24]. Alleles that reduce survival are expected to be lost at later ages. In the context of selection against hybrids, instead of testing for changes in allele frequency, researchers can test for reductions in interspecific heterozygosity (i.e., loci heterozygous for alleles that distinguish parental forms; Fig. 1b). If selection is epistatic (i.e., involves more than one locus) reductions in ancestry mismatch (i.e., non-random associations between alleles diagnostic of parental forms[25]; Fig. 1b) may also be expected.

The Swainson's thrush (*Catharus ustulatus*) provides a powerful system to identify genetic variants underlying extrinsic postzygotic isolation. Two subspecies (coastal and inland, *C. u. swainsoni* and *C. u. ustulatus*) of thrushes form a hybrid zone in western North America and take different routes on migration[26] (Fig. 1). There is strong evidence that these differences in migration reduce hybrid fitness. First, tracking data shows that hybrids take intermediate routes on migration[27]. Second, ecological modelling shows that these intermediate routes are ecologically inferior to routes taken by parental forms[28]. Third, hybrids of certain ancestries (early generation hybrids and coastal backcrosses) survive at lower rates on migration[29]. Considerable genomic work has also already been conducted in this system. For example, admixture mapping has identified genetic variants associated with migratory traits[30,31], and comparative transcriptomics has identified genes that are differentially expressed between the migratory states and subspecies[32]. Considerable misexpression has also been documented in hybrids and is more common during the migratory vs. non-migratory period[32]. Misexpression could affect hybrid fitness and may exacerbate/expose existing problems in hybrid thrushes[33].

In this work, we tested for viability selection against hybrids using large cohorts of thrushes from different age classes. We focused on changes in interspecific heterozygosity and ancestry mismatch and overlapped results with existing knowledge of migration and speciation genetics in the system. We find strong evidence for viability selection on both variables that may be connected to structural variation; in some genomic regions, selection appears to actually favor admixture in the hybrid zone between thrushes.

## Results

### Viability selection on interspecific heterozygosity at loci previously linked to migration

We captured 911 birds from the center of the hybrid zone between the coastal and inland Swainson's thrush subspecies and used plumage data to age them as hatch year (HY), second year (SY), or after second year (ASY). The earliest age class we included here (HY) has already made it through the egg, nestling, and fledgling stage; HY birds are captured right before their first migration; SY have migrated once; and ASY at least twice[34]. We genotyped birds at 2182 unlinked ancestry informative markers (loci that differentiate the subspecies) and collapsed these genotypes into a binary variable (homozygous for either coastal or inland alleles or heterozygous for inland and coastal alleles; hereafter 'interspecific heterozygosity'). Interspecific heterozygosity was used as the response variable in general linear models run for each locus, with each individual's age, genome-wide ancestry, and sex as predictor variables. We included the interaction term between age and genome-wide ancestry in these analyses as well, testing if a variant's effect on survival depends on an individual's genome-wide ancestry.

After FDR correction for multiple testing, both age and the interaction between age and genome-wide ancestry predicted interspecific heterozygosity at several loci (sex was a significant predictor at 2 loci, see Supplementary Fig. S2). Starting with age, 15.4% of variants ($n = 337/2182$) reached significance and reside on five macrochromosomes (chromosomes 1, 2, 3, 12, and Z; Fig. 2a). We include a formal analysis of overlap between these loci and former studies on migration and speciation genetics below but note here that the outlier loci on chromosome 12 were also identified as outliers in two former studies on migration genetics in the system where migratory traits were mapped[31] and a patterns of gene expression were compared between the subspecies and seasons[32]. Contrary to our expectations, however, we found an increase (rather than a decrease) in heterozygosity across age classes at loci where age predicted interspecific heterozygosity (Fig. 2b) on chromosome 12. The interaction between age and genome-wide ancestry was a significant predictor of interspecific heterozygosity at 972 loci. Two hundred and fifty of these loci occur on chromosome 5, including a region that showed an especially strong

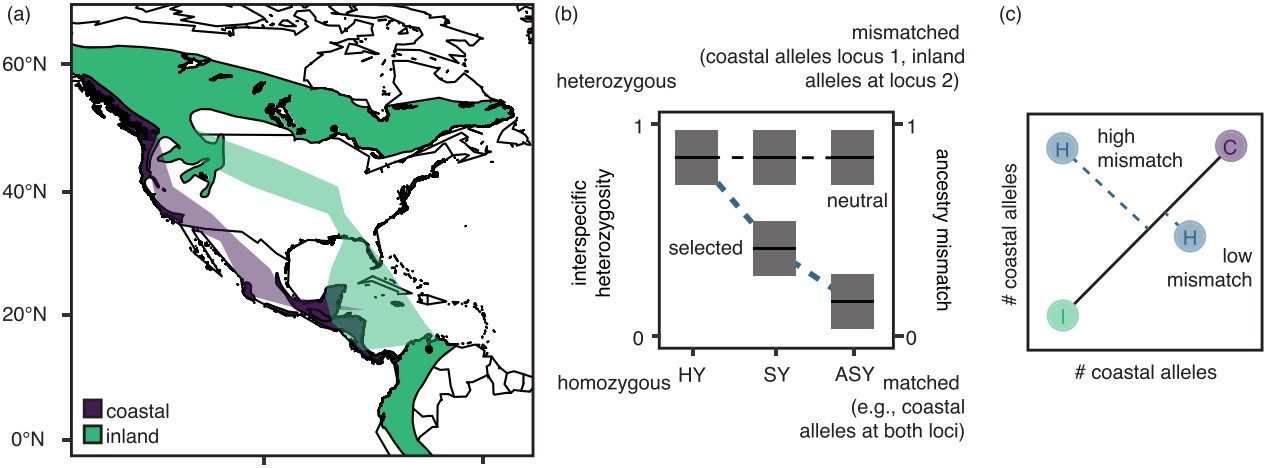

Fig. 1 | **Study system and predictions. a** Range map showing the breeding and wintering grounds for coastal (purple) and inland (green) Swainson's thrushes. Polygons in lighter color show summarized migratory routes taken by coastal (purple) and inland (green) thrushes on spring migration (Delmore et al.[26]). Range map obtained from the BirdLife International and Handbook of the Birds of the World[76]. **b** The expected results for loci under selection with a change in the frequency of interspecific heterozygosity/ancestry mismatch over time, in contrast to neutral loci. Shading reflects hypothesized prediction lines and confidence intervals. The diagram **c** visualizes how ancestry mismatch was calculated as the Euclidean distance of an individual from the line formed between the mean ancestries for coastal and inland birds. Colors reflect pure forms (purple and green) and hybrids (blue).

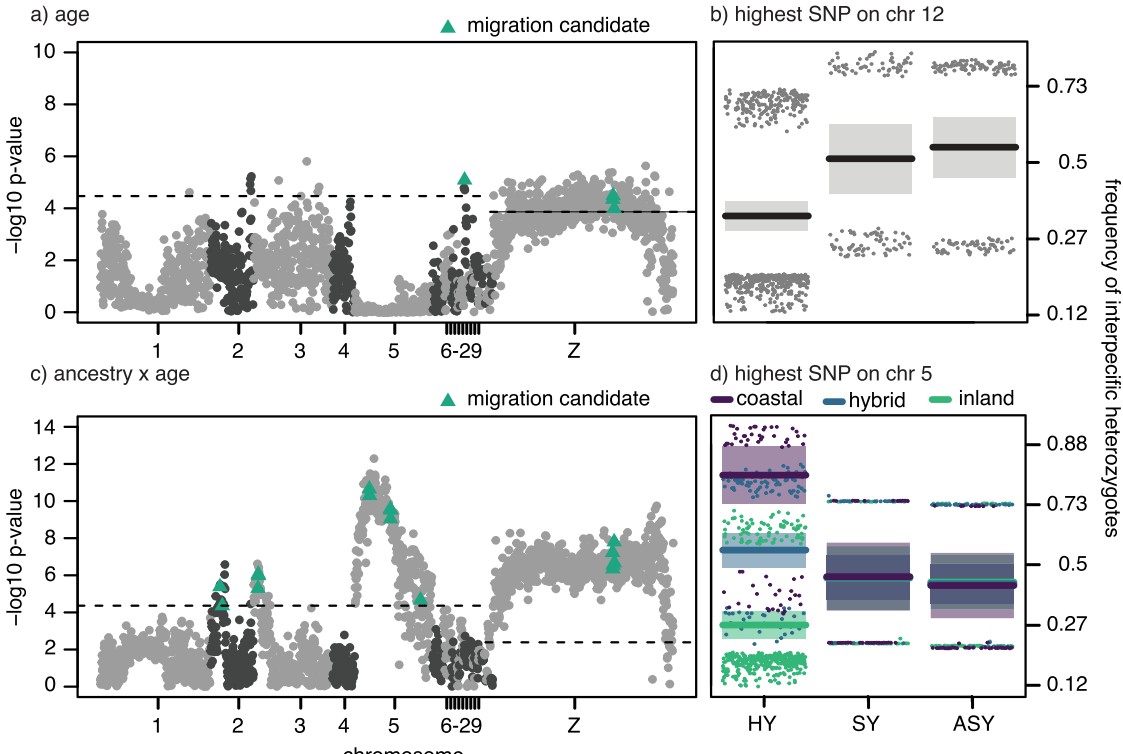

**Fig. 2 | Viability selection and interspecific heterozygosity.** Results from general linear models run for each ancestry informative marker along the genome. Results for the effects of **a**, **b** age and **c**, **d** the interaction between age and genome-wide ancestry. Separate chromosomes are shown with alternating black and grey colored points. Ancestry informative markers that include genes that were outliers in previous admixture mapping and comparative genomic analyses are indicated in green. Separate corrections for multiple testing were applied to autosomes and the Z chromosome (dotted lines indicated FDR-adjusted two-sided *p*-values < 0.05). Panels **b** and **d** show the change in interspecific heterozygosity across ages at the locus with the strongest association on chromosome 12 and 5, respectively (prediction lines and confidence intervals are shown; colors reflect birds genotyped as mostly pure [purple and green] or hybrid [blue]; see Supplement Fig. S3 for plots separated by ancestry for panel (**d**)).

association with migratory orientation in our previous genomic mapping[31]. In accordance with our expectations, birds with mostly coastal or hybrid genome-wide ancestry showed a decrease in interspecific heterozygosity across age classes at loci on this chromosome (Fig. 2d). We documented the opposite (unexpected) trend for birds with mostly inland genome-wide ancestry; in this group, there was an increase in interspecific heterozygosity across age classes. In other words, birds with mostly inland genome-wide ancestry but coastal alleles on chromosome 5 lived to later age classes. Note, many loci on chromosome 5 exhibited the same pattern, suggesting there may be strong physical linkage across this chromosome (Fig. 2c).

To conduct a more formal analysis of overlap between our results and former work on migration and speciation genetics in thrushes, we identified genes in linkage blocks around the ancestry informative loci predicted by age or the interaction between age and genome-wide ancestry above (*n* = 750 genes) and tested for overlap with genes identified in previous genomic mapping (*n* = 154 genes) and transcriptomic analyses (*n* = 234 genes; total of 358 unique genes). We found 232 genes overlapped between these lists (ancestry informative markers including those genes are shown in green, Fig. 2a, c); this number is more than the 167 genes that would be expected by chance (750/1606 genes in linkage blocks around ancestry informative markers were associated with viability and we had 358 migration candidates [or draws] [(750/1606)*358 = 167; chi-square: *p*-value < 0.0001]).

**Viability selection on ancestry mismatch at loci previously linked to migration**

The former analyses considered each locus individually. Selection may also derive from interactions between loci, especially if it stems

from differences in a complex behavior like migration that may be controlled by many loci. Accordingly, we ran a second set of analyses, testing for changes in ancestry mismatch across age classes. We predicted that selection against hybrids would remove interspecific combinations of alleles, decreasing mismatch across age classes. We estimated ancestry mismatch as the Euclidean distance between an individual's genotype at each pair of ancestry informative loci and the line between the mean ancestries for parental birds at these loci[35] (Fig. 1c) and ran the same analyses described above for interspecific heterozygosity, replacing interspecific heterozygosity with ancestry mismatch as the response variable and running separate general linear models run for each locus, with each individual's age, genome-wide ancestry, and sex, as predictor variables. The interaction term between age and genome-wide ancestry was also included.

After correcting for multiple testing using FDR, no pair of loci reached significance in tests for viability selection. Nevertheless, we noted interesting trends in uncorrected *p*-values (<0.05) for the interaction term between age and genome-wide ancestry and thus describe them here (sex was not a significant predictor at any pairs of loci). Large numbers of variants on a subset of macrochromosomes (4390/2,379,891 pairs of loci tested; on chromosomes 1, 2, 3, 5, and Z) exhibited changes in ancestry mismatch across age classes (Fig. 3a, c). Changes in ancestry mismatch between variants on chromosomes 1 and Z; and chromosomes 5 and Z were especially prominent (Fig. 3a, c) and are interesting in the context of thrushes as variants linked to migration mapped to all three chromosomes in our previous work[30,31]. Following our expectations, the ancestry mismatch between loci on both chromosomes 1 and Z and chromosomes 5 and Z decreased for

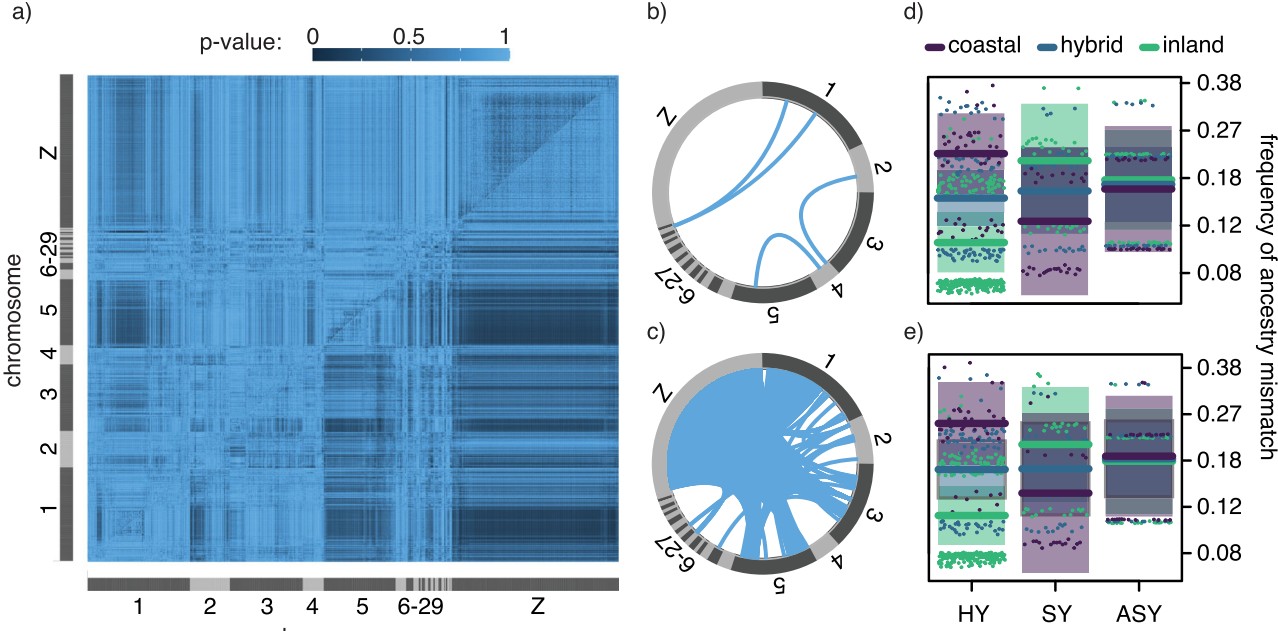

**Fig. 3 | Viability selection and ancestry mismatch.** Results from general linear models run for each pair of ancestry informative markers present in the genome. **a** *p*-Values for the effects of age (top triangle) and the interaction between age and genome-wide ancestry (bottom triangle; blue shading reflects unadjusted two-sided *p*-values). **b, c** Pairs of loci that are located on different chromosomes and for which models in **a** returned a *p*-value < 0.05 for **b** age and **c** the interaction between age and genome-wide ancestry. **d, e** The change in ancestry mismatch across ages at the loci with the strongest association between chromosome 1 and the Z (**d**) and chromosome 5 and the Z (**e**) (predictions lines and confidence intervals are shown; colors reflect birds genotyped as mostly pure [purple and green] or hybrid [blue]; see Supplement Fig. S4 for plots separated by ancestry).

birds with both mostly coastal and hybrid (to a lesser degree) genome-wide ancestry. Similar to results for interspecific heterozygosity, that was not the case for birds with mostly inland genome-wide ancestry; ancestry mismatch increased across age classes for this group and for loci on both chromosomes 1 and Z and chromosomes 5 and Z. Also similar to results for interspecific heterozygosity, large blocks of loci on each chromosome exhibited the same pattern, suggesting these loci are in strong physical linkage.

**Structural variants, including at least one pericentric inversion, encompass loci under viability selection**

Our analyses of viability selection showed that blocks of variants often exhibit the same patterns on chromosomes 1, 5 and Z, suggesting that loci on these chromosomes are physically linked and/or in areas of reduced recombination. We expect reductions in recombination on the Z chromosome as it only recombines in males (the homogametic sex in birds). Whereas on the autosomes, structural variants (SVs), like inversions, may contribute to reductions in recombination[36]. We investigated the connection with SVs further here, starting by estimating recombination rates across these chromosomes using whole-genome resequencing data from parental inland birds. We documented significantly lower recombination in the same areas that show evidence of viability selection compared to the rest of the genome (Wilcoxon rank sum test: $W = 242,755$, *p*-value < 0.001). This pattern holds for both regions on chromosomes 1 (between 25 and 100 Mb) and 5 (between 20 and 60 Mb; Fig. 4a).

Next, we used principal component analyses to estimate local population structure along chromosomes 1 and 5. SVs like inversions should create three clusters corresponding to each inversion genotype (homozygous coastal, heterozygous, and homozygous inland), and elevated heterozygosity is expected in the central (heterozygous) cluster[37,38]. In accordance with these expectations, outlier regions with three clusters were identified on both chromosomes (encompassing 21.6 Mb on chromosome 1, between 40 and 61 Mb, and 14.8 Mb on chromosome 5, between 21 and 36 Mb) and heterozygosity was elevated in the central clusters in both cases (Fig. 4c). These patterns are consistent with two clusters of individuals that are homozygous for alternative, putative inversion haplotypes and an intermediate cluster of individuals that are heterozygous for the inversion haplotype with very little recombination between them. We confirmed this finding using a second analysis, estimating linkage disequilibrium (LD; Fig. 4b) in (1) all samples (bottom triangle) and (2) only samples homozygous for the more common haplotype based on our PCA analyses (top triangle). We expected to find elevated LD across all samples but not in homozygotes, but this pattern was mostly limited to chromosome 5. Elevated LD was also documented in a small region on chromosome 1, but found in both heterozygotes and homozygotes (Fig. 4b).

In our final analysis, we used Hi-C data to compare the structural organization of coastal and inland genomes. Specifically, we used Hi-C data for one individual from each subspecies to generate interaction matrices aligned to the inland genome for each subspecies. We then estimated the interaction differences between the subspecies to identify long-distance interactions present only in the coastal subspecies. Focusing on the outlier regions identified in local PCAs and analyses of LD, we documented increased long-distance interactions on chromosome 5 (at 28 × 18 Mb and 38 × 18 Mb) but not chromosome 1 (Fig. 4d). Interestingly, interaction plots generated for each subspecies separately show that centromeres occur in our candidate regions on both chromosomes (~50 Mb on chromosome 1 and ~28 Mb on chromosome 5; Supplementary Fig. S7).

## Discussion

Seasonal migration is emerging as a compelling example of an extrinsic postzygotic isolating barrier[39,40]. The migratory divide between Swainson's thrushes exemplifies this form of isolation, with previous work showing hybrids navigate over areas that are ecologically inferior to those of parental forms[28] and survive migration at

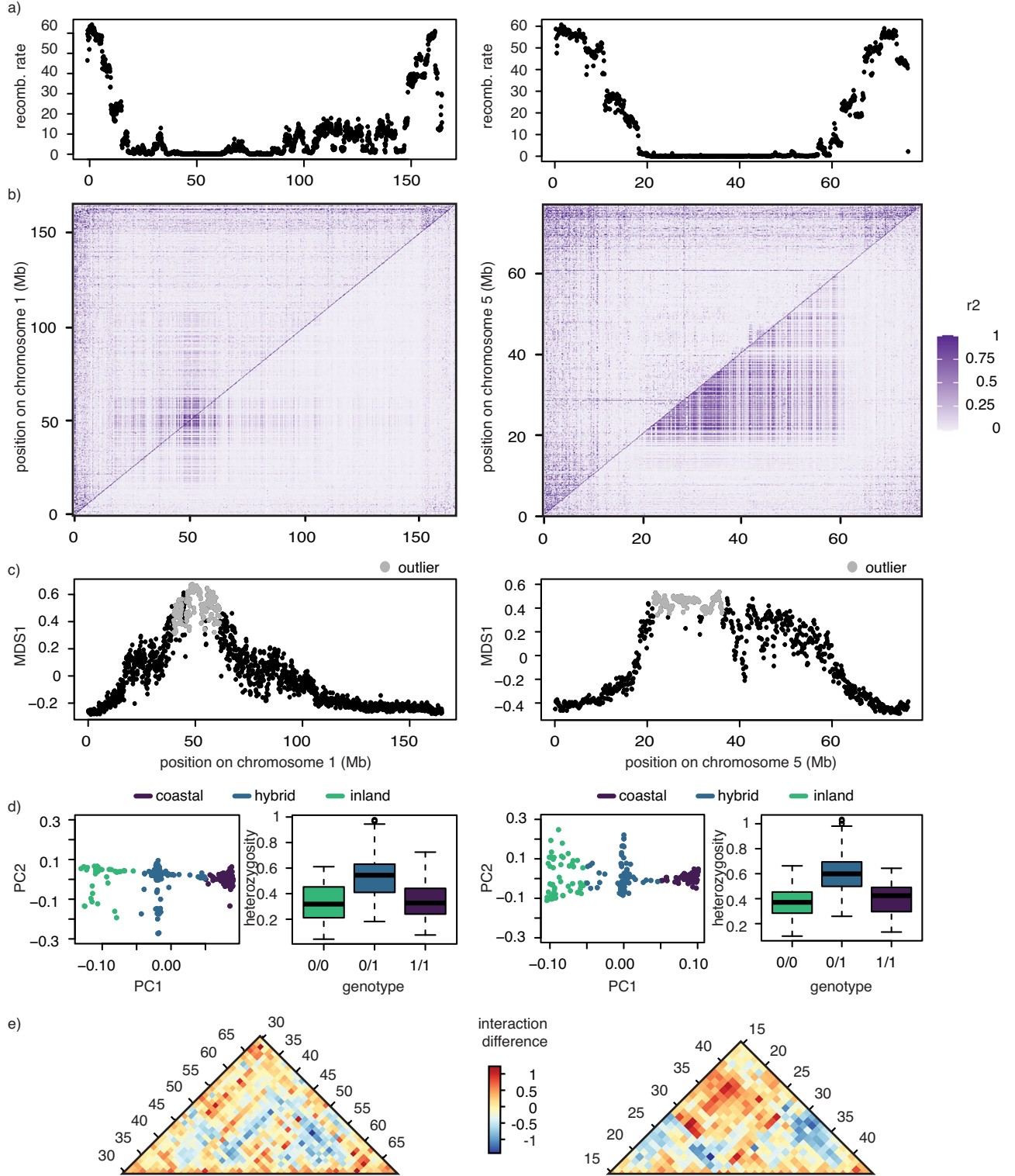

**Fig. 4 | Structural variants under viability selection.** Series of analyses testing for evidence of structural variants on chromosomes 1 (left) and 5 (right), including **a** estimates of recombination rates, **b** linkage disequilibrium in all individuals (bottom) or individuals homozygous (top) for putative variants (purple shading reflects the strength of r2 [linkage disequilibrium]), **c** local PCA analyses (100 kb windows with outlier population structure shown in gray), **d** PCA on outlier windows and levels of heterozygosity in the PCA clusters (boxplots show minimum, maximum, median, first and third quartiles; colors reflect birds genotyped as pure [purple or green] or hybrid at these loci; n = 296 birds, including those from our largest hybrid populations and pure, allopatric populations), and **e** Hi−C link pattern between inland and coastal genomes (red indicates elevated long-distance interactions; blue decreased long-distance interactions).

lower rates[29]. We find additional support for migration's role as an extrinsic postzygotic isolating barrier in thrushes here, with regions of the genome that control migratory traits exhibiting signatures of viability selection in hybrids. We expand on these findings below, including a discussion of the unexpected result that some hybrids may actually be more fit than parental forms in the hybrid zone. We also expand on evidence that epistatic interactions and structural variants contribute to the reduced fitness of hybrid thrushes.

## Hybridization as a creative force in evolution

The sorting and recombining of parental alleles in hybrids may have allowed a subset of thrushes to occupy one or more underutilized adaptive peaks within the species. Specifically, age predicted interspecific heterozygosity at a subset of loci on chromosomes 5 and 12, but instead of reductions in heterozygosity, this variable increased across age classes (in the case of chromosome 5, this increase was limited to birds with mostly inland genome-wide ancestry). These patterns indicate that birds heterozygous for coastal and inland alleles at these genomic regions are favored by selection. This finding could be explained by the traits encoded by these genomic regions. Specifically, migratory distance and orientation map to these regions, with coastal alleles coding for shorter, more western routes[31]. The inland subspecies of Swainson's thrush migrates over longer distances than the coastal subspecies, navigating to eastern North America before heading south to winter in South America[26]. Coastal birds head directly south and winter closer to their breeding grounds, in Mexico and Central America[26]. Seasonal migration is an energetically costly behavior[41,42] that is characterized by high mortality rates[43]. Shorter, more western routes could allow inland birds with coastal alleles at these regions to survive at higher rates. Note, only a small number of genes occur in the region on chromosome 12, and these include acetylcholine receptors; genetic variation at these genes alone could reduce fitness costs associated with migration (e.g., acetylcholine receptors have been connected to increased endurance and cognitive performance[44,45]).

Considering these results from a broader perspective, they align with the combinatorial view of speciation that hybridization can facilitate the reassembly of old genetic variants into new combinations and can thus facilitate the origin of novel phenotypes with a fitness advantage[46]. In the context of Swainson's thrushes, hybridization with the coastal form may have allowed a subset of inland birds to jump the fitness valley between an adaptive peak well-suited to traditional inland migratory routes (hypothesized to reflect the routes they used to expand out of eastern refugia following the last glacial maximum[47]) and a peak better suited to inland birds that breed in western (vs. eastern) North America.

## Epistatic selection in ecological speciation

The results we present here are also pertinent to discussions about the role of epistatic interactions in speciation. Specifically, age alone did not predict interspecific heterozygosity on chromosome 5; it was the interaction between age and genome-wide ancestry that predicted this variable. This indicates that only a subset of hybrids are being selected against and helping maintain reproductive isolation in the system (i.e., those with mostly coastal or hybrid genome-wide ancestry but interspecific heterozygosity on chromosome 5). While not significant following corrections for multiple testing, we also documented interesting trends in ancestry mismatch between loci on chromosomes 1, 5 and Z that were also dependent on genome-wide ancestry, implicating both local (e.g., between specific loci on chromosomes 1 and 5) and more genome-wide epistatic interactions affecting hybrid fitness. Discussions of the role epistatic interactions play in speciation are often limited to intrinsic selection against hybrids, with alleles that evolved at separate loci during allopatry being incompatible in hybrids upon secondary contact[48], but there is growing evidence that divergent natural selection can also generate epistatic interactions with extrinsic effects on hybrid fitness[13].

In the context of seasonal migration and thrushes, migration is a complex behavior that integrates many traits (e.g., the timing, distance, and orientation of migratory routes, morphology of wings, and timing of physiological traits like molt and hyperphagia[49,50]). Inland and coastal subspecies have diverged in many of these traits[26,51,52] and a mismatch across these traits could affect an individual's ability to migrate successfully. We documented a mismatch at genomic regions

that code several of these traits; we already mentioned that orientation maps to chromosome 5. Wing morphology maps to chromosome 1, and several traits within the migratory syndrome map to the Z chromosome. Future work could use machine learning approaches to test if interactions between multiple traits and genomic regions can predict survival in the system; machine learning approaches are suitable for this kind of high-dimensional dataset, where nonlinear, complex relationships likely exist[53,54].

We have focused on the role the genetic variants we identified here could play in extrinsic selection against hybrids, as previous work in the system suggests this is the main form of selection maintaining stability in the hybrid zone[27,28,32] and we found overlap with loci previously connected to migration. Intrinsic selection is also thought to take quite some time to evolve in birds[55]. Nevertheless, we want to highlight here that the loci we identified could also underlie other forms of selection. Results for the Z chromosome exemplify this point, as sex chromosomes are known to play a disproportionate role in speciation independent of migration[56,57]. For example, in the context of epistatic interactions, recessive alleles will be exposed in the heterogametic sex and can lead to incompatibilities with other loci in the genome[56,58]. Evolution is also thought to be faster on the sex chromosomes (e.g., because effective population sizes are lower on these chromosomes, facilitating genetic drift and/or because alleles involved in sexual selection and conflict often reside on these chromosomes and can be involved in a runaway selective process[59]). Accordingly, faster evolution on the Z could result in even more genetic incompatibilities in our system.

## Structural variants and centromeres in evolution

Our results also contribute to growing evidence that structural variants and centromeres may facilitate speciation, as blocks of loci on chromosomes 1 and 5 showed similar patterns of viability selection. These regions exhibited both reduced recombination and local population structure, patterns indicative of structural variants[60,61]. A comparison of LD in all birds included in the study to birds homozygous in these regions and interaction plots generated by Hi–C data allowed greater resolution into what these variants might be on chromosome 5. Specifically, LD was only elevated in heterozygotes on this chromosome and stronger long-distance interactions in the coastal, but not inland, subspecies were documented with the Hi–C data. Both patterns are consistent with an inversion between 18 and 38 Mb.

We did not document any evidence for large SVs, like inversions, on chromosome 1; patterns of LD did not contrast between all birds and birds homozygous for putative SVs in the region, and long-distance interactions were not observed in the Hi–C data. Instead, consistent patterns of viability selection on this chromosome may reflect the action of many small structural variants, acting together to reduce recombination and generate local population structure. The presence of so many breakpoints and variants may obscure patterns in both the Hi–C interaction and LD plots. The suggestion that several small SVs may occur in this region on chromosome 1 is supported by previous work using short read data to call structural variants on this chromosome, with many small SVs, including deletions, insertions and inversions occurring here[31].

One common finding for both chromosomes 1 and 5 was that centromeres occur in both regions (indicated by subspecies-specific Hi–C interaction plots). Similar associations were recently documented in deer mice[62] and could derive from the fact that centromeres often harbor highly repetitive sequences that can facilitate ectopic recombination and the formation of structural variants[63]. Note, centromeres on their own can also reduce recombination, and that appears to be the case on chromosome 1, where LD is elevated in both all birds and birds homozygous for the putative SVs. It is possible that the centromere on this chromosome has catalyzed linked selection in the region on its own (i.e., independent of SVs). Future work using

long-read data to call SVs on both chromosomes will provide greater resolution into the role SVs play in this system.

Overall, our results suggest that hybridization and migration can have variable effects on the fitness of hybrids in migratory divides. A subset of hybrid thrushes appears to have found an unoccupied adaptive peak in the system, exemplifying the creative role hybridization can have in evolution. This finding does not argue against the role differences in seasonal migration could still play in reducing hybrid fitness and maintaining reproductive isolation in divides, as another subset of hybrids still appear to be selected against (those that are mainly coastal or hybrid with admixture on chromosomes 1, 5 and Z). Epistatic interactions, structural variants and centromeres appear to be influencing the evolutionary trajectories each group of hybrids is taking. Beyond these findings, we hope our analyses have highlighted the power of viability selection analyses. We look forward to future work using similar approaches to broaden our understanding of ecological speciation and other evolutionary processes. This work should incorporate the whole genome, considering the genomic background of each individual and interactions with other unlinked loci in the genome.

## Methods

All sampling was performed in accordance with relevant guidelines and regulations, including permits obtained from Environment and Climate Change Canada (10921 and 10921A), the State of Alaska Department of Fish and Game (20-1134; 21-117; 22-083), the Washington Department of Fish and Wildlife (20-104; 21-076; 22-061), United States Geological Survey (24199), United States Fish and Wildlife Service (MB65923D) and the United States Department of Agriculture (137701).

### Sampling

We used mist nets and audio recordings of Swainson's thrush song to capture birds ($n = 911$) in the hybrid zone between coastal and inland Swainson's thrush in Hope (49.385, −121.316), Pemberton (50.264, −122.867; British Columbia, Canada), Hyder (55.950, −130.039; Alaska, USA) and Cle Elum (47.323, −121.099, Washington, USA) between June and September 2010–2013 and 2019–2023 (note, capture year had no effect on downstream analysis, see Supplementary Fig. S5). We aged birds as hatch year (HY), second year (SY) or after second year (ASY) using plumage and molt limits in the field[34] and took blood (~25 μl) or feather (fourth retrice) samples for genomic analyses.

### DNA extraction and sequencing

We obtained blood samples from most birds ($n = 846$) and used a standard phenol–chloroform protocol for DNA extraction. We only obtained feather samples for a subset of birds (mostly adults fitted with geolocators in years 2010–2013 and not recaptured the following year, $n = 65$). We extracted DNA from these feathers using a digestion solution (1 M Tris; 1 M NaCl; 0.1 M CaCl$_2$; 10% SDS; 1 M DTT and H2O), proteinase K and the QIAGEN PCR purification kit to precipitate and clean the DNA.

All individuals included in this study were sequenced and genotyped following methods described in Justen et al.[31]. Starting with tagmentation, 2–5 nanograms of DNA was added to a mix of TDE1 Illumina Buffer, homemade buffer (20 mM Tris-Hcl, 10 mM MgCl$_2$) and Tn5 transposase enzyme (TDE1–Illumina) pre-charged with custom adapters and incubated at 55 °C for 5 min. One of 96 custom indices (i7) was added to each sample on a plate in addition to a mastermix including an i5 index and OneTaq HS Quick-Load 2×. The PCR reaction included denaturation at 95 °C, annealing at 55 °C and extension at 68 °C; 12 PCR cycles were used. After amplification, 10 μl of each individual reaction was pooled and purified using AMPure XP beads (1×). Library size distribution and quality were visualized on the Bioanalyzer 1000 (Agilent, Molecular Genomics Workspace, Texas A&M University) and size selected between 350 and 750 bp. Libraries were sequenced on the Illumina NovaSeq 6000 to 1–13× of coverage (median 3.55×; pair-end 150 bp reads).

Low-quality sequences were removed after alignment of sequences to reference genomes using BWA. We called an initial set of SNPs using bcftools (--min-BQ 20, --min-MQ 20, %QUAL > 500, --skip-variants indels) and imputed missing genotypes with STITCH[64], a program that uses a hidden Markov model to estimate individuals' haplotype probabilities. We ran STITCH in 1 Mb blocks (with a buffer of 100 kb), initiated the program in the pseudoHaploid model (80 and 500 for K−ancestral haplotypes and nGen−number of generations since population was founded) and switched to the diploid model after 36 EM. The final set of genotypes will be referred to as the whole-genome sequencing dataset−hereafter WGS.

### Ancestry informative markers

We adapted a pipeline from Schumer et al.[65] to identify ancestry informative markers and genotype birds (i.e., assign ancestry states) at these markers (for main steps of the pipeline see Supplementary Fig. S1). First, we identified a set of putative ancestry informative markers using existing, de novo reference genomes for both the coastal and inland subspecies of thrushes. We made these genomes colinear by simulating fastq files (reads) from the coastal reference genome (fasta file) using wgsim and aligning the resulting fastq files to the inland reference genome with bwa. We called variants in the coastal reference sequence using bcftools and filtered these variants for quality (QUAL < 30 || DP < 7 || DP > 60 || MQ < 40). We used bcftools to get a consensus sequence for the coastal genome, giving us a coastal pseudoreference genome that is colinear to the inland reference genome. We extracted all sites with fixed differences between the collinear reference genomes as an initial set of putative ancestry informative markers.

Next, we used a reference panel of high coverage (average read depth 18× whole genome resequencing data from 28 birds (14 from each subspecies, birds captured in populations allopatric to the hybrid zone; sequencing and analysis of high-coverage data is described in Louder et al.[32]) to determine a final set of ancestry informative markers. We aligned the reference panel reads to both the inland and coastal reference genome assemblies with bwa, retaining reads that mapped to both, and called variants using bcftools. We then extracted the set of putative ancestry informative markers. Using the reference panel, we estimated allele frequency differences between the subspecies at each putative ancestry informative marker, retaining sites with an allele frequency difference greater than 0.5 to produce a final set of ancestry informative markers. We then estimated allele counts for each subspecies at each ancestry informative marker.

We called ancestry informative markers from low coverage genomes of hybrids with ancestryHMM[65,66], which uses a hidden Markov model (HMM) to infer ancestry as a hidden state based on observations of sequencing read counts, given reference species allele counts and the genetic distance between loci. We estimated read counts at ancestry informative markers for low coverage individuals by first aligning each sequence to both the inland and coastal reference genomes with BWA, retaining only reads that mapped to both. We called variants and extracted ancestry informative markers with bcftools, then extracted read counts from the resulting VCF files. We then estimated the genetic distance between ancestry informative markers in cMs using a recombination map estimated with LDhat[67] (https://github.com/auton1/LDhat) and the physical distance between sites. We ran ancestryHMM with starting parameters of one ancestry pulse 6000 generations ago and 77% average genome-wide inland ancestry (average genome-wide ancestry in the population is 77% and we used these parameters: -a 2 0.77 0.23 -p 0 −1000000 0.77 -p 1 −3000 0.23). The model estimated ancestry informative markers at 643,414 loci across the genome (-640 markers/Mb). We used vcftools to remove loci with more than 75% missing data for our dataset and further performed a filtering through LD pruning of ancestry informative markers (--indep-pairwise 200 20 0.2 −maf 0.05), resulting in 2184 loci for

downstream analysis. For downstream analysis, we converted posterior probabilities (>0.9) at each ancestry informative site to hard genotype calls. Sites with a greater than 0.9 posterior probability for any ancestry state were assigned to that state, while sites <0.9 posterior probability were converted to NAs. Homozygous inland, heterozygous, and homozygous coastal ancestry calls were assigned genotype states of 0–2, respectively. We used these ancestry informative marker loci to calculate genome-wide ancestry (calculated by the mean ancestry informative marker value across loci) and heterozygosity (calculated by the proportion of 1 s at ancestry informative marker loci in the dataset). For example, a parental coastal bird would have an ancestry of 0, while a F1 hybrid would have an ancestry of 0.5 and an inland bird an ancestry of 1.

### Viability selection and interspecific heterozygosity

We estimated interspecific heterozygosity at each ancestry informative marker by collapsing genotypes into a binary variable (homozygous for either coastal or inland alleles or heterozygous for inland and coastal alleles; hereafter, interspecific heterozygosity). We tested for viability selection using generalized linear models with interspecific heterozygosity as the response variable and each individual's age (ordinal variable, HY, SY, or ASY), genome-wide ancestry (continuous variable, 0–1), and sex as predictor variables. We also included the interaction between genome-wide ancestry and age in the models, because previous work in the system has shown that birds with different ancestry experience differences in selection[29] and to test for potential epistatic interactions between heterozygosity at a specific locus and ancestry at the rest of the genome [glm(heterozygosity at ancestry informative marker locus- genome-wide ancestry * age + sex, family = binomial(link = "logit"))]. Once we identified outlier loci (after FDR adjustment; $p$-value < 0.05), we estimated linkage blocks (LD; $r^2 > 0.5$) around these loci, extracted genes from within the LD blocks that we consider as being under viability selection (750 genes; see 'overlap with previous genomic analyses' below for further details).

### Ancestry mismatch

We compared ancestry at all pairwise loci combinations of ancestry informative markers across the genome. For each individual, we estimated the mismatch between each pair of loci as the Euclidean distance of an individual from the line formed between the mean ancestries for parental birds at these loci, following Chhina et al.[35]. Low values of ancestry mismatch indicate more similar ancestry at the two loci, while high ancestry mismatch values indicate different ancestry. We used ancestry mismatch as the response variable in generalized linear models that we ran for each pairwise locus pair separately. Similar to glms for our analysis of interspecific heterozygosity, we ran the models using genome-wide ancestry, age, sex, and the interaction between genome-wide ancestry and age as covariates [glm(ancestry mismatch at loci pair - genome-wide ancestry * age + sex, family = poisson(link = "log"))]. No glms showed significantly associated loci pairs after adjusting for multiple testing (FDR adjustment; p-value < 0.05). We plotted results from these models using R packages 'ggplot2' and 'visreg'[68,69]. We additionally plotted loci pairs between chromosomes for which the unadjusted $p$-value was below 0.05 in a circular plot using the R package 'BioCircos'[70].

### Overlap with previous genomic analyses

We estimated linkage disequilibrium around all ancestry informative markers in our dataset using our WGS dataset and vcftools (--geno-r2 positions). We used these estimates to identify intervals (linkage blocks, all loci with $r^2 > 0.5$) around each ancestry informative marker and extracted genes from within these LD blocks ($n = 1606$ genes). We then tested for overlap between genes in linkage blocks around loci significantly predicted in our analyses and two previous analyses in the system. In Justen et al.[31], we used tracking data from wild birds and whole genome resequencing data to map a series of migratory traits, including migratory timing, orientation and distance, as well as wing morphology (192 candidate genes for all migratory traits). Of these genes, 154 were associated with ancestry informative markers used in the present study (Supplementary Data 1; 9.6%). In Louder et al.[32], we used a captive population of thrushes and transcriptomic data to identify genes differentially expressed between the migratory states (wintering vs. spring migration, $n = 300$), the interaction between subspecies and migratory states (genotype × environment, $n = 488$) and genes that were misexpressed in hybrids ($n = 1942$). This work was conducted in five different brain regions that likely play an important role in migration. A total of 234 genes identified in the transcriptomic analysis were associated with ancestry informative markers used in the present study (Supplementary Data 1; 14.6%). We calculated number of genes that were expected to overlap by chance by dividing the number of genes associated with viability by the total number of genes multiplied by the number of migration candidates [(750/1606) * 358 = 167]). We calculated $p$-values for the overlap using a chi-square test applying the 'allwise_p_chisq' function in R[71].

### Recombination

We estimate recombination across the genome using high coverage, short-read data from parental populations and LDhat to estimate recombination rate using 'interval' to generate a likelihood file 'rho-map' (-its 1000000 -samp 2000 -bpen 5.0 -burn 2000 -exact, additional details for this approach can be found elsewhere[31]). Recombination rate estimates were summarized into 100 kb windows. We used these estimates to test if loci identified as outliers in our viability analysis are more often found in windows of low recombination compared to loci not connected to viability. We first extracted estimates of recombination from windows for each ancestry informative marker based on their location in the genome. Second, we compared medians between groups (outlier or non-outlier in our viability analysis using interspecific heterozygosity) using an unpaired two-sample Wilcoxon rank sum test (as data were not normally distributed).

### Linkage disequilibrium plot

To characterize potential structural variants identified in previous analyses further, we computed LD across chromosomes 1 and 5 using two subsample groups of the data. Subset (1) all samples in the population, including heterozygotes, and (2) only the samples homozygous for the more common haplotype (i.e., inland haplotype). We used vcftools (VCFtools/0.1.16) to filter the WGS data for SNPs with a minor allele frequency >5% (--maf 0.05), number of missing genotypes (--max-missing 0.1) and thinned data to include one SNP per 10 kb (--thin 10000; chromosome 5) and per 100 kb (--thin 100000; chromosome 1). Filtered data was subsetted by sample group and chromosome. We used plink (PLINK/1.9b5) to calculate $r^2$ for pairwise loci in the dataset (--ld-window 100000000 --ld-window-kb 999999999 --ld-window-r2 0) and plotted results using the R package 'ggplot2'[69].

### Local PCA

To explore structural variants further, we performed local PCAs with the lostruct package[61] in R using WGS data of birds from our largest hybrid populations (Hope and Pemberton; $n = 268$) and our reference panel of birds from pure, allopatric populations ($n = 14$/subspecies). We followed the approach described in Harringmeyer & Hoekstra[62] for this analysis. Specifically, we ran local PCAs in 100 kb windows (step size = 100 kb) and used the pc-function with default parameters to calculate the distance between PCA clusters (using the top two PCs). To visualize distances between clusters, we used multidimensional scaling (MDS) with the cmdscale function. To identify outlier regions as candidates for structural variants, we scanned for neighboring 100 kb windows that showed similar population structure to each

other, but distinct population structure from the rest of the chromosome. We first performed a k-means clustering analysis of the 100 kb windows using two MDS axes and chose the k with the maximum silhouette score (best $k = 2$ for chr 5 and $k = 4$ for chr 1; see Supplementary Fig. S6). After assigning the 100 kb windows to either cluster 1 or 2, we calculated z scores for MDS1 scores for each 100 kb window (using the mean and standard deviation for all MDS1 scores). Outlier regions were defined by z-score > 1 and a minimum of ten consecutive windows next to each other.

## PCA and heterozygosity

Principal component analyses (PCA) can also be used to identify structural variants (inversions should create three clusters corresponding to each inversion genotype (homozygous coastal, heterozygous, and homozygous inland) and elevated heterozygosity is expected in the central (heterozygous) cluster[37,38]. Accordingly, we subset our WGS data to regions identified as outlier regions by local PCAs using vcftools (VCFtools/0.1.16) and subsequently performed PCAs using plink (PLINK/1.9b5) for the entire outlier region. We visualized PC1 and PC2 and assigned 3 clusters. We compared genome-wide heterozygosity (based on ancestry informative markers genotypes) of individuals in each cluster and plotted results in basic R.

## Hi−C

We used interspecific comparisons of Hi−C matrices as an additional approach to identify structural variants, following Todesco et al. [72]. Hi−C sequencing was performed for the inland and coastal reference genomes when they were constructed (Louder et al.[32], Vertebrate Genomes Project 2021). We aligned each fastq file to the inland reference genome using bwa mem with the following parameters: '-A1 -B4 -E50 -L0'[73]. We then sorted and indexed the resulting bam files with samtools[74]. With hicFindRestSite, from 'HiCExplorer', we identified the locations of the restriction enzyme cut site sequences ('GATC' for the coastal Hi−C sequencing and 'GA.TC' for the inland Hi-C sequencing) in the inland genome[75]. We then used hicBuilMatrix to construct interaction matrices for each of the coastal and inland sequences, with a bin size of 10 kb. We then normalized the two matrices to the same read coverage, with hicNormalize, then corrected each matrix with z-score thresholds of −1.5 and 5. We then combined bins with hicMergeMatrixBins to produce 1 Mb bins. To compare chromatin interactions between the coastal and inland genomes, we used hicCompareMatrices with the operation 'log2ratio'. Note, it is possible that our use of the inland genome for alignment biased this analysis to identifying more long-distance interactions in the coastal subspecies, but we think it is unlikely, given the size of regions showing these interactions (much larger than individual short reads).

## Reporting summary

Further information on research design is available in the Nature Portfolio Reporting Summary linked to this article.

# Data availability

Sequencing data of all individuals used in this study is made available on the NCBI Sequence Read Archive under BioProject numbers PRJNA979932, PRJNA1024534, and PRJNA1157204. Distribution map obtained with permission from the BirdLife International and Handbook of the Birds of the World[76] (http://datazone.birdlife.org/species/requestdis). Source data used to generate figures are provided in the Source Data file or uploaded to Dryad under https://doi.org/10.5061/dryad.pzgmsbd11. Source data are provided with this paper.

# Code availability

A github repository including bioinformatic pipelines used in this study is available following this link: https://github.com/HannahJusten/swth_vaibility_selection.

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

## Acknowledgements
This research was supported by an NSF CAREER (IOS-2143004) and NIH (1R35GM151012) grant to KED and several additional grants to HCJ (an AAUW International Fellowship and research grants through the American Philosophical Society, Schubot Center for Avian Health at Texas A&M University, and the Animal Behavior Society). We thank members of the Delmore lab for field assistance. We thank Rachel Moran for comments on an earlier version of this paper.

## Author contributions
H.C.J. and K.E.D. conceived and designed this work; H.C.J. and K.E.D. collected the data; H.C.J. and S.A.B. conducted the analyses with assistance from K.E.D.; H.C.J. and K.E.D. wrote the paper with revisions from S.A.B.

## Competing interests
The authors declare no competing interests.
