## [Transparent Peer Review file · Nature Communications]

The genetics of extrinsic postzygotic selection in a migratory divide between subspecies of the Swainson's thrush

Corresponding Author: Dr Hannah Justen

Version 0:

Reviewer comments:

Reviewer #1

(Remarks to the Author)

This is overall a very interesting and strong manuscript investigating genetic signatures of extrinsic selection against hybrid Swainson's thrushes. I have a few questions and suggestions for strengthening what is already a strong paper.

1. A range map, perhaps as a Supplemental Figure, would help clarify the study system and the divergence between the two subspecies' migratory routes (Lines 59-63, and line 80).
2. Lines 69-71: By "misexpression is more common on migration", do you mean temporally, such that more misexpression happens while the thrushes are migrating? I think this was just a minor clarification issue with the language.
3. First two results subheadings note selection on loci linked to migration. Aren't the identified loci associated with the two different subspecies more generally, as opposed to migratory behavior per se? If I am wrong, this link to migration needs to be stated much more explicitly. If I am correct, then I think these two headings should be edited so that they don't overstate the link with migratory behavior.
4. Lines 133-136: I had difficulty following this explanation of how ancestry mismatch was calculated. Could this be clarified? Would a diagram be possible?
5. Lines 146-152: In the presentation of the ancestry mismatch results here, results are presented only for birds with coastal or inland genome-wide ancestry. But if there is ancestry mismatch, then these birds must have at least some mixed/hybrid ancestry, correct? Is it just such a small amount--so many generations of backcrossing?--that it is not detectable at the genome-wide level? Or is this analysis of ancestry mismatch really getting at degree of hybrid ancestry, and I misunderstood that it was being done in birds that are mostly in one ancestry group? Clearly I was a bit confused here.
6. Lines 434-436 and 451-452: Why did the general linear models not include sex in any interaction terms? Was this a power issue? Was there a biological rationale? It would be helpful for the chosen model structure to be explained explicitly.

Minor typos

Line 97: studies instead of studying

Line 116: informative instead of informatic

Line 120: Shouldn't this be Fig 1a,c instead of 1bc?

Line 143: Figure 2a,c instead of 2a,b

Line 144: Figure 2a,c instead of 2a,b

Figure 2 legend, lines 735 and 736: Shouldn't this be "top triangle" and "bottom triangle", instead of "top of the triangle", etc.?

Reviewer #2

(Remarks to the Author)

I have reviewed the manuscript "The genetics of extrinsic postzygotic selection in a migratory divide between songbirds" by

Hannah Justen, Stephanie Blain and Kira Delmore. The authors present interesting and thorough analyses of viability selection in the Swainson's Thrush. They find reductions in interspecific heterozygosity and ancestry mismatch with age in some birds, although this pattern depends on genome-wide ancestry. These results suggest selection against a subset of hybrids, whereas other hybrids are potentially more fit than the parental taxa. Moreover, the viability selection could be traced to putative structural variants in the genome. These are fascinating findings that highlight an aspect of speciation research that has received less attention, namely the genetic underpinnings of extrinsic postzygotic selection.

The manuscript is clearly structured and nicely written, and the analyses have been performed properly (although I have added a few suggestions in the minor comments below). More importantly, I think that the manuscript can be improved in terms of clarity. Specifically, there is no clear definition of viability selection that explains the underlying processes and expected genetic patterns. It would be helpful for the reader if the authors could provide more details on expected results (perhaps by including an additional figure in the introduction). Also, the main figures need more attention. Although all information is included, they could be presented more clearly. I have made some suggestions below. Overall, I have no major concerns with this work and hope to see it published quickly. It is an important addition in the literature on avian speciation and hybridization.

Minor comments

Line 15: Could you provide more information or some examples of how seasonal migration can lead to extrinsic isolation?

Line 16: Mention the species name.

Line 20-21: Provide more details about the supporting evidence. It remains quite vague in the abstract.

Line 48: Could you mention a few processes unrelated to selection that might affect these analyses?

Line 49-52: Although you explain viability selection at the end of this sentence (i.e. changes in allele frequencies across age classes), it would be helpful to provide a clear definition. It is now implicitly explained in the text.

Line 53-57: These processes (reduction in interspecific heterozygosity and ancestry mismatches) need more explanation.

How do these genetic changes occur under viability selection, and which patterns do you expect to see in the analyses. This additional information will help the reader to understand the subsequent analyses and results.

Line 59-60: Add scientific names of species and subspecies.

Line 73-74: Perhaps briefly describe the expected results.

Line 93 & 138: How did you correct for multiple testing?

Line 96-100: This explanation is difficult to follow without a brief recap of the previous studies. A few extra sentences would be helpful to guide the reader through this section.

Line 101: Unclear what "this locus" refers to.

Line 120-124: This explanation is unclear. Can you rephrase it?

Line 133-136 & 444-446: Unclear what you mean with "the line between mean ancestries" here. And even though you performed similar analyses as for interspecific heterozygosity, I would recommend to briefly repeat structure of the linear models.

Line 164-166: Would it be possible to directly connect the recombination rate with the results from the previous section, for example by correlating recombination with interspecific heterozygosity and ancestry mismatches?

Line 178-183: The explanation of the results is clear, but it is difficult to link the described patterns to the figures. Perhaps add an extra (supplementary) figure to illustrate the LD-patterns in homozygotes and heterozygotes.

Line 187: Could the alignment to an inland genome introduce a bias in this analysis?

Line 227-235: Not completely clear which message you want to convey here. Could you rephrase and/or restructure this paragraph?

Line 235-243: Interesting perspective! For clarity, I would make this into a separate paragraph.

Line 269: Why machine learning approaches? Could you provide more details about this suggestion for further research, and add some supporting references?

Line 273-286: How would you disentangle intrinsic and extrinsic selection on the Z-chromosome? Could you already indicate which type of selection is more prevalent in your study system?

Line 352-353: Could there be an effect of catch period (2010-2013 vs. 2019-2020)? You could add catch period as a random factor to the linear models to account for this variation.

Line 370-373: This procedure needs more explanation. Which setting did you use?

Line 377: Before detailed explanations of the different steps in this pipeline, it would be helpful to provide a short overview of the main steps. This could potentially be visualized in a supplementary figure.

Line 380: What do you mean with "simulating fastq files"?

Line 411: Why 77% average genome-wide inland ancestry? What is the reasoning behind this threshold?

Line 434 & 450: I noticed that the effect of sex is not mentioned in the results. Did sex have any effect on interspecific heterozygosity or ancestry mismatch? It would be good to briefly mention this in the results section.

Line 468 & 474: Perhaps add percentages to the number of genes that overlapped with previous studies.

Line 521-525: Briefly explain the purpose of this analysis.

Figure 1: Add y-axis below the top-panel of figure 1a (even though it is present in the lower part). Panel 1d is very difficult to read and understand. Perhaps split it up into three separate sections?

Figure 2a: Add legend and more information on axes.

Figure 2b,c: Not very readable. Is it necessary to show this, or could you visualize it in a different way?

Figure 2d,e: Too much data in one graph. I recommend to split it up.

Figure 3e: Add explanation for the legend and the axes.

Version 1:

Reviewer comments:

Reviewer #1

(Remarks to the Author)

The authors have done a thorough job responding to the comments from both reviewers. I found one small typo in the Figure 1 legend on line 781: "neural" should be "neutral."

I am satisfied with the revision of this interesting manuscript.

Reviewer #2

(Remarks to the Author)

I have reviewed the revised manuscript "The genetics of extrinsic postzygotic selection in a migratory divide between songbirds" by Hannah Justen, Stephanie Blain and Kira Delmore. I want to applaud the authors on a great revision. They have nicely incorporated my suggestions or clearly justified when they did not follow my comments. I especially appreciate the addition of Figure 1 which nicely shows the expected results. I only have a few minor comments that can be easily addressed.

Line 18: Perhaps also add the scientific name in the abstract.

Line 49-52: Nice that you provided some additional explanation on other processes unrelated to selection. However, it still remains quite vague (e.g., what do you mean with "genomic features"). I would suggest adding a few concrete examples.

Line 52: It is now clear what viability selection entails. However, the definition between parentheses (locus-specific changes in allele frequency) could be more precise. For example, by also referring to different age classes.

Line 79: The introduction adds quite abruptly. You could add a few sentences where you recap the expected patterns, or already provide a summary of the main results.

Line 98 & 146: Mention that the correction for multiple testing was FDR.

Line 106-108: Indicate that these patterns were found on chromosome 12. Now they read as general patterns.

Line 173-176: I would suggest to slightly rephrase this sentence. The current version give the impression that you correlated recombination rate with viability selection (which would require a Spearman correlation). However, you compared recombination rates between two groups of genomic regions (which indeed requires a Wilcoxon test). Perhaps write "significantly lower recombination". Additionally, report the p-value as "p-value < 0.001". Finally, replace "is true" with "holds".

Line 422: For clarity, I would report the information between parentheses as a separate sentence.

Line 505: I believe that a Wilcoxon test compares medians, not means.

Figures 2 & 3: I really appreciate the addition of supplementary figures S3 and S4 to show the separate plots. Personally, I prefer the supplementary figures because they show the results more clearly. But I leave the final choice with the authors.

Figure S3: Typo in the middle panel (should be "hybrid").

We thank the reviewers for the time they spent on our manuscript, their comments improved our manuscript substantially. Responses to each comment are shown in bold, along with line numbers where changes to the manuscript were made. Note, line numbers refer to the track changes version of the manuscript we uploaded.

Reviewer #1 (Remarks to the Author):

This is overall a very interesting and strong manuscript investigating genetic signatures of extrinsic selection against hybrid Swainson's thrushes. I have a few questions and suggestions for strengthening what is already a strong paper.

Thank you for your kind remarks. We are happy you see the strength of this manuscript.

1. A range map, perhaps as a Supplemental Figure, would help clarify the study system and the divergence between the two subspecies' migratory routes (Lines 59-63, and line 80).

Thank you for this suggestion. We have added a range map that includes polygons summarizing migratory routes taken by both subspecies to the main text (Figure 1) and refer to it in line 65.

2. Lines 69-71: By "misexpression is more common on migration", do you mean temporally, such that more misexpression happens while the thrushes are migrating? I think this was just a minor clarification issue with the language.

This was from a comparison of misexpression in captive birds that were assayed during both the migratory and non-migratory period. We found greater levels of misexpression during the migratory period. We have added these details (lines 74-76).

3. First two results subheadings note selection on loci linked to migration. Aren't the identified loci associated with the two different subspecies more generally, as opposed to migratory behavior per se? If I am wrong, this link to migration needs to be stated much more explicitly. If I am correct, then I think these two headings should be edited so that they don't overstate the link with migratory behavior.

Thank you for noting this lack of clarity. We are referring to the overlap analyses outlined at the end of each section, where we test if loci previously linked to migration (through both admixture mapping in Justen et al. 2024 PNAS and comparative gene

expression analyses in Louder et al. 2025 Nat Comm) occur in the same regions that show evidence of viability selection in the present study. We have added the word “previously” to each subheading (lines 83 and 133). Hopefully this makes the connection clearer.

4. Lines 133-136: I had difficulty following this explanation of how ancestry mismatch was calculated. Could this be clarified? Would a diagram be possible?

Thank you for this suggestion. We have added a diagram to explain the calculation of mismatch to the new Figure 1 (c) in the main text.

5. Lines 146-152: In the presentation of the ancestry mismatch results here, results are presented only for birds with coastal or inland genome-wide ancestry. But if there is ancestry mismatch, then these birds must have at least some mixed/hybrid ancestry, correct? Is it just such a small amount--so many generations of backcrossing?--that it is not detectable at the genome-wide level? Or is this analysis of ancestry mismatch really getting at degree of hybrid ancestry, and I misunderstood that it was being done in birds that are mostly in one ancestry group? Clearly I was a bit confused here.

Thank you for noting the absence of this detail. We have added information for hybrids, noting that they also show the expected trend of reduced ancestry mismatch across age classes (lines 158-159).

6. Lines 434-436 and 451-452: Why did the general linear models not include sex in any interaction terms? Was this a power issue? Was there a biological rationale? It would be helpful for the chosen model structure to be explained explicitly.

Thank you for this question. Sex was included in the analyses. We have added a plot to the Supplementary Information showing those results (Figure S2) and outlined the results between lines 99-100 and 151. We did not have an a priori reason to expect a three way interaction between sex, genome-wide ancestry and age so we did not include this term in the models. We added information to explain the chosen model structure in lines 457-461.

Following this comment, we have rerun the models including a three way interaction (`glm(interspecific heterozygosity~genomewide-ancestry*age*sex, family=binomial(link="logit"), data=data frame)`). This term does not predict interspecific heterozygosity (see figure below); to keep things simple and in line with our a priori expectations we have left this analysis out.

interaction: ancestry * age * sex

Results from general linear models run for each ancestry informative marker along the genome. Results shown for the interaction between genome-wide ancestry, age and sex. Different chromosomes shown by different color of the points.

Minor typos

Line 97: studies instead of studying

Change made line 103.

Line 116: informative instead of informatic

Change made line 123.

Line 120: Shouldn't this be Fig 1a,c instead of 1bc?

Change made line 127.

Line 143: Figure 2a,c instead of 2a,b

Change made line 153.

Line 144: Figure 2a,c instead of 2a,b

Change made line 155.

Figure 2 legend, lines 735 and 736: Shouldn't this be "top triangle" and "bottom triangle", instead of "top of the triangle", etc.?

Changes made lines 799 and 800.

Reviewer #2 (Remarks to the Author):

I have reviewed the manuscript “The genetics of extrinsic postzygotic selection in a migratory divide between songbirds” by Hannah Justen, Stephanie Blain and Kira Delmore. The authors present interesting and thorough analyses of viability selection in the Swainson’s Thrush. They find reductions in interspecific heterozygosity and ancestry mismatch with age in some birds, although this pattern depends on genome-wide ancestry. These results suggest selection against a subset of hybrids, whereas other hybrids are potentially more fit than the parental taxa. Moreover, the viability selection could be traced to putative structural variants in the genome. These are fascinating findings that highlight an aspect of speciation research that has received less attention, namely the genetic underpinnings of extrinsic postzygotic selection.

Thank you. We are glad you agree that our findings will help fill a knowledge gap in speciation genetics.

The manuscript is clearly structured and nicely written, and the analyses have been performed properly (although I have added a few suggestions in the minor comments below). More importantly, I think that the manuscript can be improved in terms of clarity. Specifically, there is no clear definition of viability selection that explains the underlying processes and expected genetic patterns. It would be helpful for the reader if the authors could provide more details on expected results (perhaps by including an additional figure in the introduction). Also, the main figures need more attention. Although all information is included, they could be presented more clearly. I have made some suggestions below. Overall, I have no major concerns with this work and hope to see it published quickly. It is an important addition in the literature on avian speciation and hybridization.

Thank you for your feedback. We have addressed each individual comment below and added a figure to the main text outlining our predictions as suggested above (Figure 1b).

Minor comments

Line 15: Could you provide more information or some examples of how seasonal migration can lead to extrinsic isolation?

Thank you for this suggestion. We have added an example here (lines 16-17) and note we expand on this hypothesis in the introduction (starting line 65).

Line 16: Mention the species name.

Change made line 18.

Line 20-21: Provide more details about the supporting evidence. It remains quite vague in the abstract.

Thank you for this suggestion. We define viability selection a few sentences above as “locus-specific changes in interspecific heterozygosity and ancestry mismatch across age classes”. We hope that is sufficient for the abstract given word constraints.

Line 48: Could you mention a few processes unrelated to selection that might affect these analyses?

Thanks for this suggestion; this will increase clarity for readers. We have made the requested change lines 50-51.

Line 49-52: Although you explain viability selection at the end of this sentence (i.e. changes in allele frequencies across age classes), it would be helpful to provide a clear definition. It is now implicitly explained in the text.

We have made the requested change (line 52-53) and as suggested in your initial remarks, we have added a figure with our predictions for these analyses (Figure 1b).

Line 53-57: These processes (reduction in interspecific heterozygosity and ancestry mismatches) need more explanation. How do these genetic changes occur under viability selection, and which patterns do you expect to see in the analyses. This additional information will help the reader to understand the subsequent analyses and results. **Hopefully the new figure we have added helps clarify these questions (Figure 1).**

Line 59-60: Add scientific names of species and subspecies.

Additions made lines 62 and 64.

Line 73-74: Perhaps briefly describe the expected results.

Hopefully the new figure we have added helps clarify these questions (Figure 1b) and referred to in lines 58 and 60.

Line 93 & 138: How did you correct for multiple testing?

FDR adjustment. These details can be found in the Methods lines 463 and 479.

Line 96-100: This explanation is difficult to follow without a brief recap of the previous studies. A few extra sentences would be helpful to guide the reader through this section.

Thank you for this suggestion. We have added details (lines 105-107).

Line 101: Unclear what “this locus” refers to.

Modified to “loci where age predicted interspecific heterozygosity” (line 108-109).

Line 120-124: This explanation is unclear. Can you rephrase it?

We have rephrased as requested (lines 127-131).

Line 133-136 & 444-446: Unclear what you mean with “the line between mean ancestries” here. And even though you performed similar analyses as for interspecific heterozygosity, I would recommend to briefly repeat structure of the linear models.

We have repeated the structure of linear models as requested (lines 143-146) and have added a panel to Figure 1c describing how ancestry mismatch was estimated. Thank you for this suggestion, we think it will improve the clarity of our analyses.

Line 164-166: Would it be possible to directly connect the recombination rate with the results from the previous section, for example by correlating recombination with interspecific heterozygosity and ancestry mismatches?

Thank you for this suggestion. We ran the suggested analysis to compare recombination rate between significant outlier identified in our study and non-outlier loci. We focused on results from our analysis of interspecific heterozygosity for this, since the analysis of ancestry mismatch did not recover significant associations. Following our expectation, we found lower values of recombination in significant outliers (mean $\rho=0.25$) compared to non-outlier loci (mean $\rho= 2.13$). The difference between groups is statistically significant, we have added a note about these results to the Results (lines 175-177) and Methods (lines 513-519).

Line 178-183: The explanation of the results is clear, but it is difficult to link the described patterns to the figures. Perhaps add an extra (supplementary) figure to illustrate the LD-patterns in homozygotes and heterozygotes.

We have added additional details to the description that should help connect what we have written to the figure (lines 193-194). Hopefully the addition of Figure 1 showing the predictions and how ancestry mismatch was estimated will also help clarify this section.

Line 187: Could the alignment to an inland genome introduce a bias in this analysis?

Interesting question. If we understand the reviewer's comment, they are suggesting that reads from the coastal subspecies will be missed or reported incorrectly if they are too distinct from the inland genome. If these reads are aligned elsewhere in the genome this could generate signatures of long-distance interactions. We think this possibility is unlikely; we observe a cluster of 1 Mb windows showing elevated signatures of long distance interaction between windows at ~19 and ~40 Mb. Given that our HiC sequencing data is comprised of short reads, the presence of the pattern in multiple windows of this size suggests that it is not produced by reads that overlap one region erroneously mapping to another. Instead, it indicates that there is a consistent pattern of long distance interaction in the coastal HiC data in that region that is not present in the inland HiC data. In addition, the interaction plot we have generated with the HiC data is only one of several lines of evidence showing that an inversion or complex set of inversions occurs in this region. Regardless, we appreciate the reviewers' comment and have added a note of caution to the manuscript (lines 576-579).

Line 227-235: Not completely clear which message you want to convey here. Could you rephrase and/or restructure this paragraph?

We have removed some of this content, appending the note about genes to the previous paragraph and (as requested below), splitting the last part off into its own paragraph (lines 241-246).

Line 235-243: Interesting perspective! For clarity, I would make this into a separate paragraph.

Thanks for this suggestion. We have made it into a separate paragraph (lines 251-258)

Line 269: Why machine learning approaches? Could you provide more details about this suggestion for further research, and add some supporting references?

Relationships between traits in the migratory syndrome and genomic regions that underlie them and selection against hybrids are likely complex and non-linear. Machine learning approaches are well suited to this kind of high dimensional data analysis. We have added these details and references to the document (lines 286-288).

Line 273-286: How would you disentangle intrinsic and extrinsic selection on the Z-chromosome? Could you already indicate which type of selection is more prevalent in your study system?

We are not aware of any genomic approach that can be used to test if selection derives from intrinsic vs extrinsic selection. In some cases, authors can use the identity of loci under selection and complementary experimental work to make this distinction (e.g., Powell et al. 2020 Science 368) but this would be difficult in our system; the entire Z chromosome shows very similar patterns making it difficult to identify causal loci and experimental work is difficult with wild birds. We are happy to employ an approach if the reviewer is familiar with one. Given the length of our discussion, we have chosen to leave this content out for now.

Line 352-353: Could there be an effect of catch period (2010-2013 vs. 2019-2020)? You could add catch period as a random factor to the linear models to account for this variation.

Thanks for this suggestion. We have rerun the models adding release year as a co-variable. We did not recover any significant associations (figure below). In addition, the results reported for additional predictors did not change. Since we have no a priori reason to expect catch period to affect our results we have chosen to leave the models as they were in the main manuscript. We added results from this iteration of the models in the Supplementary Figure S5 and refer to it in the main text (line 370).

Results from general linear models run for each ancestry informative marker along the genome. Results shown for the co-variable release year. Different chromosomes shown by different color of the points.

Line 370-373: This procedure needs more explanation. Which setting did you use?

We have added additional detail to this section in lines 390-394.

Line 377: Before detailed explanations of the different steps in this pipeline, it would be helpful to provide a short overview of the main steps. This could potentially be visualized in a supplementary figure.

Thank you for this suggestion, we have added a figure to the Supplementary Information as suggested (see Figure S1, lines 399-400).

Line 380: What do you mean with “simulating fastq files”?

Create reads from the genome fasta file. We have added this detail on line 403 and in Figure S1.

Line 411: Why 77% average genome-wide inland ancestry? What is the reasoning behind this threshold?

That is the average genome-wide ancestry in our population. This detail has been added line 434.

Line 434 & 450: I noticed that the effect of sex is not mentioned in the results. Did sex have any effect on interspecific heterozygosity or ancestry mismatch? It would be good to briefly mention this in the results section.

Thank you for noting this fact. Sex was a significant predictor at two loci for interspecific heterozygosity (and no pairs of loci in the case of ancestry mismatch). We have added a note to the paper (lines 99-100; 151) and present results for this variable in the Supplementary Information (Figure S2).

Line 468 & 474: Perhaps add percentages to the number of genes that overlapped with previous studies.

Percentages have been added (lines 495, 501).

Line 521-525: Briefly explain the purpose of this analysis.

Purpose added lines 553-556.

Figure 1: Add y-axis below the top-panel of figure 1a (even though it is present in the lower part). Panel 1d is very difficult to read and understand. Perhaps split it up into three separate sections?

Thank you for this feedback. We have added a figure to the Supplementary Information (Figure S3) that splits panel 1d by ancestry as suggested and refer to these plots in lines 794-795.

Figure 2a: Add legend and more information on axes.

We made modifications to the figures as requested.

Figure 2b,c: Not very readable. Is it necessary to show this, or could you visualize it in a different way?

These figures are a second way of visualizing results in Figure 2a. We think these panels may actually be more readable as they are only showing loci with p-values <0.05 from panel a. Accordingly, we prefer to leave these panels in.

Figure 2d,e: Too much data in one graph. I recommend to split it up.

Thanks for this suggestion. We have added a figure to the Supplementary Information splitting the data by genome-wide ancestry (Figure S4, lines 804-805).

Figure 3e: Add explanation for the legend and the axes.

Modification made.

Response to reviewers

Reviewer #1 (Remarks to the Author):

The authors have done a thorough job responding to the comments from both reviewers. I found one small typo in the Figure 1 legend on line 781: "neural" should be "neutral."

I am satisfied with the revision of this interesting manuscript.

Thank you for reviewing the manuscript a second time. We have corrected the typo and appreciate your support with this second revision.

Reviewer #2 (Remarks to the Author):

I have reviewed the revised manuscript "The genetics of extrinsic postzygotic selection in a migratory divide between songbirds" by Hannah Justen, Stephanie Blain and Kira Delmore. I want to applaud the authors on a great revision. They have nicely incorporated my suggestions or clearly justified when they did not follow my comments. I especially appreciate the addition of Figure 1 which nicely shows the expected results. I only have a few minor comments that can be easily addressed.

Line 18: Perhaps also add the scientific name in the abstract.

Added (line 17).

Line 49-52: Nice that you provided some additional explanation on other processes unrelated to selection. However, it still remains quite vague (e.g., what do you mean with "genomic features"). I would suggest adding a few concrete examples.

Additional detail added concerning genomic features that reduce recombination (lines 48-49).

Line 52: It is now clear what viability selection entails. However, the definition between parentheses (locus-specific changes in allele frequency) could be more precise. For example, by also referring to different age classes.

Thank you. We have added age classes to this definition (lines 51-52).

Line 79: The introduction adds quite abruptly. You could add a few sentences where you recap the expected patterns, or already provide a summary of the main results.

Thank you. We have added an additional sentence summarizing the main results (lines 79-81).

Line 98 & 146: Mention that the correction for multiple testing was FDR.

Detail added (lines 100 and 148).

Line 106-108: Indicate that these patterns were found on chromosome 12. Now they read as general patterns.

Detail added (line 110).

Line 173-176: I would suggest to slightly rephrase this sentence. The current version gives the impression that you correlated recombination rate with viability selection (which would require a Spearman correlation). However, you compared recombination rates between two groups of genomic regions (which indeed requires a Wilcoxon test). Perhaps write “significantly lower recombination”. Additionally, report the p-value as “p-value < 0.001”. Finally, replace “is true” with “holds”.

Suggested changes made (lines 175-177).

Line 422: For clarity, I would report the information between parentheses as a separate sentence.

We have kept as is. This detail is better suited to parentheses.

Line 505: I believe that a Wilcoxon test compares medians, not means.

Thank you. Change made (line 515).

Figures 2 & 3: I really appreciate the addition of supplementary figures S3 and S4 to show the separate plots. Personally, I prefer the supplementary figures because they show the results more clearly. But I leave the final choice with the authors.

We have kept the figure as is. We believe the single panel should be a sufficient summary of trends.

Figure S3: Typo in the middle panel (should be “hybrid”).

Change made.